# A Middle-Range Theory of Social Isolation in Chronic Illness

**DOI:** 10.3390/ijerph20064940

**Published:** 2023-03-10

**Authors:** Paolo Iovino, Ercole Vellone, Nadia Cedrone, Barbara Riegel

**Affiliations:** 1Department of Health Sciences, University of Florence, 50134 Florence, Italy; 2Department of Biomedicine and Prevention, University of Rome Tor Vergata, 00133 Rome, Italy; 3Department of Nursing and Obstetrics, Wroclaw Medical University, 50-367 Wroclaw, Poland; 4Unità di Medicina Interna, Ospedale S. Pertini, 00157 Rome, Italy; 5School of Nursing, University of Pennsylvania, Philadelphia, PA 19104, USA; 6Mary MacKillop Institute for Health Research, Australian Catholic University, Fitzroy, VIC 3065, Australia; 7Center for Home Care Policy & Research, VNS Health, New York, NY 10017, USA

**Keywords:** chronic disease, social isolation, social interaction, loneliness, nursing theory

## Abstract

Chronic illnesses and social isolation are major public phenomena that drive health and social policy worldwide. This article describes a middle-range theory of social isolation as experienced by chronically ill individuals. Key concepts include social disconnectedness, loneliness, and chronic illness. Antecedents of social isolation include predisposing factors (e.g., ageism and immigration) and precipitating factors (e.g., stigma and grief). Outcomes of social isolation include psychosocial responses (e.g., depression and quality of life), health-related behaviors (i.e., self-care), and clinical responses (e.g., cognitive function and health service use). Possible patterns of social isolation in chronic illness are described.

## 1. Background

Social isolation is a powerful determinant of poor health, with a significant impact on morbidity and mortality in populations worldwide [1,2]. In adults, about one in ten individuals experience social isolation, with sociodemographic and socioeconomic factors influencing prevalence [3].

The high prevalence of chronic illnesses has increased the risk of social isolation [4]. Given that chronic illnesses are more prevalent in middle-aged and older individuals, this population is particularly susceptible to social isolation during their disease trajectory [5]. Moreover, the impact of isolation on individuals with chronic illness is generally worse than that of a healthy population [4]. Specifically, the onset of social isolation in people with a chronic illness is complex; health problems can alter one’s social network; for example, these individuals may view themselves as different from healthy persons due to disabling symptoms and related discomfort, or struggle to engage in social activities due to a lack of energy. As a result, the ill person may lack emotional support and experience loneliness [6].

Social isolation was also greatly exacerbated by the Sars COVID-19 pandemic. Due to the need to curb mortality and morbidity caused by this infection, governments around the world have been urged to take extreme restrictive measures, such as home isolation, and quarantine. Not only has this intensified loneliness [7], but also disrupted chronic care due to the postponement of scheduled medical visits, and delayed care seeking [8].

Although there is ample literature that describe the phenomenon of social isolation across the world [9], the knowledge on how this process is engendered in chronically ill individuals remains understudied. Middle-range theory can be derived from grand theory, developed inductively from qualitative research, or derived through logical analysis and synthesis [10]. This middle-range theory was developed deductively through an extensive review of theoretical and empirical literature, with the goal of explaining the onset and outcomes of social isolation in chronic illness.

Middle-range theories are essential in nursing research because they strengthen the scientific base of the nursing discipline and are close enough to observed data to permit the incorporation of propositions for empirical testing, thus guiding clinical practice [10]. In this paper, we outline the building blocks of the theory, including concepts, assumptions, propositions, and the logic of the phenomenon of interest. The implications for nursing practice and research are discussed.

## 2. Operational Definition of Concepts

The core concepts of this middle-range theory are social disconnectedness, loneliness, and chronic illness. The first two concepts are often studied in tandem in empirical research and are embedded under the umbrella term social isolation. The term social isolation captures a dense, multi-dimensional construct, reflecting the structural and functional aspects of social engagement or relationships [11]. Chronic illness represents the context in which the phenomenon of social isolation is described in this middle-range theory.

### 2.1. Social Disconnectedness

The seminal work of Cornwell and Waite (2009) defines social disconnectedness as an objective measure of social isolation that reflects physical separation from other individuals. Drawing on the indicators collected by the National Social Life, Health, and Aging Project (NSHAP) [12], this theory considers disconnectedness as a composite of the domains of social network characteristics, living arrangements, number of friends and family members, and degree of social participation (Table 1).

A person affected by a chronic illness is at risk of experiences that lead to alterations in indicators of objective social disconnectedness. For example, the level of engagement allowed by the illness can compromise social contacts. Second, the persons with whom the ill individual shares particular activities may withdraw because they can no longer share them. These alterations are particularly problematic because chronic illness may impose a greater need for social support [13]. 

### 2.2. Loneliness

Loneliness is defined as an unpleasant subjective experience in response to social isolation, determined by the perception of a discrepancy between the relationships one expects and the objective relationships one has [14]. This theory adopts the operationalization theory proposed by Weis (1973) [15], in which loneliness is described in terms of its emotional and social dimensions. Emotional loneliness is the perceived lack of an attachment figure and someone to turn to (e.g., a partner or a best friend), while social loneliness refers to the absence of a broader network of friends and other acquaintances that can provide a sense of belonging, companionship, and feelings of being a member of a community. An ever-expanding body of literature indicates that people with chronic illnesses are more predisposed to feelings of loneliness than healthy individuals [5], as described below.

### 2.3. Chronic Illness

In this theory, we adopt the term chronic illness, defined as a multidimensional construct that captures not only the presence of a long-term biomedical alteration, but also the individual experience of living with a chronic disease. Experiences are referred to as the psychosocial aspects that a chronic illness engenders, such as social isolation and social stigma.

## 3. Antecedents of Social Isolation

Several factors can increase the likelihood of developing social isolation in people with a chronic illness. In this theory, we classify them as predisposing and precipitating factors (Figure 1).

### 3.1. Predisposing Factors

The predisposing factors to social isolation are defined as preexisting conditions and include age, gender, immigration status, occupational status, living environment, sexual orientation, personality traits, and genetic predisposition. Abundant evidence indicates that social isolation is relatively more frequent in older adults [16]. The main reasons for this higher prevalence can be attributed to an intrusive illness that affects activities of daily living, retirement, or the loss of loved ones (e.g., spouse, family member, or friends) [17]. As described in the subsequent paragraphs, age indirectly affects social isolation via other factors (e.g., ageism).

Social isolation also varies according to gender; in fact, women have been found to have broader and stronger social networks than men [18]. Another important predisposing factor is immigration status; evidence suggests that immigrants are more predisposed to social isolation than non-immigrants, probably linked to stressors such as language barriers and differences in cultural background [19]. Importantly, this population is also more likely to be exposed to discrimination and racism, which trigger personal insecurity towards social interactions and social participation [20]. Racism is a problem not confined to immigration; extant literature suggests that this phenomenon also affects White individuals, with serious consequences, including emotional reactions and feelings of loneliness [21].

Employment can protect against loneliness [22] because it requires less self-directed effort to remain socially engaged. Low income is another predisposing factor to loneliness. Cohen-Mansfield, Shmotkin, and Goldberg (2009) [23] prospectively studied older people and found after 3.5 years that loneliness occurred mainly in those reporting limited financial resources, probably due to the imposed limits on specific leisure activities (e.g., trips, outgoings and hosting friends at one’s residence).

Environmental factors can also facilitate social interaction; in general, living in an urban rather than a rural location is favorable, due to the greater availability of social resources, whereas the neighborhood crime level negatively impacts social interactions due to the perceived threats to safety [24]. However, different living environments can impact the lives of people affected by chronic illnesses; for example, a specific health issue can exacerbate feelings of living in a high-crime neighborhood, making them more reluctant to leave their residence. Another factor that exacerbates social isolation may be no longer driving due to a decline in physical health. This can be an important issue for people who live in places with few transportation options [16]. Another factor that influences social isolation is the healthcare environment itself; for example, long-term care residences can increase or decrease isolation, depending on factors such as the provision of home-like accommodations, ease of contact with family and friends, presence of technology, and comfortable private spaces [25].

There is evidence that different personality traits predict social isolation. For example, Iveniuk (2019) [26] found that extraverted and agreeable people had larger and stronger social network ties than their counterparts; however, other personality characteristics (i.e., conscientiousness) can be positive in relation to social network outcomes [27].

### 3.2. Precipitating Factors

Precipitating factors are risk factors that, in clusters or alone, trigger the onset of loneliness or social disconnectedness. Many of these precipitating factors are unrelated to chronic illness, but they make coping with an illness relatively more challenging, such as the loss of a significant social network member. Other precipitating factors are directly related to chronic illness, reflecting the extent of the physical and psychological intrusiveness of the illness itself. These factors include stigma, grief, the frequency, severity, and bothersomeness of symptoms, physical dysfunction, sensory deficits, body image changes, lack of self-esteem, low sense of belonging, and poor quality social support, as discussed below.

Stigma across individuals with chronic illness represents a growing area of research; evidence in this field suggests the high complexity of this construct across those living with invisible illnesses. This group can experience stigma in the forms of anticipated (i.e., expectations of stigma experiences in the future), internalized (feelings of self-directed prejudice caused by absorbing negative stereotypes from society), and enacted stigma (i.e., the experience of unfair behavior perpetrated by others) [28]. The characteristic of invisibility of chronic illness offers the key to explain the possible dynamics of isolation onset; firstly, the society can act by discrediting and devaluing the person, as a result of perceiving their symptoms as “exaggerating”; secondly, the chronically ill individuals may react by starting to adopt coping secrecy and social withdrawal. This further reinforces internalized stigma, thus perpetrating stigma-related social isolation [29]. A more subtle but frequent form of stigmatization is also represented by the experience of pity or compassion conveyed by family members and friends [30].

Ageism is another precipitating factor for isolation, given that it is relatively similar to stigma. Ageism is defined as the negative stereotypes, prejudices and discriminations toward old age and the aging process. Although it is still a relatively understudied concept, ageism can be an important risk factor for late-life loneliness, through a mechanism of social rejection (e.g., mandatory retirement) and stereotype embodiment (i.e., negative self-perception of aging due to stereotypes) [31]. The resultant isolation can be worsened in the presence of an intrusive illness that leads to one’s deterioration of physical health (e.g., compromised mobility).

Grief is an emotional reaction commonly associated with chronic illness. A recent overview describes grief as an adjustment process complicated by the flare-ups of symptoms, progression and incurability of the illness, and related impairments. The consequences of an active grieving state are hostility, low self-esteem, and self-isolation [32].

Low self-esteem is common in patients with chronic illness [33]. Low self-esteem is related to negative social comparisons, feelings of inadequacy, and excessive self-criticism. Furthermore, some chronic illnesses lead to alterations in body image (e.g., obesity, psoriasis and mastectomy with breast cancer) and physical function. The emotional reaction resulting from these illnesses, together with the anticipated stigma, triggers a progressive decline in self-image and self-esteem [34], which becomes the basis for exclusion from a range of daily social interactions.

A sense of belonging, defined as the extent to which an individual feels connected to and part of the social community [35], can deteriorate with chronic illness due to experiences of social detachment, self-blame, alienation, and social stigma [36]. In this situation, the ill individual may have many contacts and experience interactions, but does not feel part of the community.

Finally, a lack of emotional and instrumental social support can be considered as a precipitating factor because chronically ill people (especially older adults) rely heavily on family members and friends to cope with their health problems (i.e., informal caregivers). The Salutogenic Model posits that social support is particularly important in boosting generalized resistance resources and adaptively coping in stressful situations [37]. Social interactions and relationships are a source of emotional and instrumental social support. In conditions with physical and psychosocial needs, such as at the onset or during the exacerbation of a chronic illness, reciprocal communication and tangible help are fundamental. Lack of perceived support (e.g., from an intimate caregiver or another family member) is traumatizing for chronically ill people due to the ensuing unmet needs, which can precipitate feelings of loneliness and depression and worsen physical health [38].

It is important to emphasize that precipitating factors can be multiple or recurrent events that trigger the onset of social isolation, especially when they coexist with the predisposing conditions. For example, a person living in a rural area can work to preserve social connections despite a scarce social network. However, if the illness has made the person feel vulnerable or fragile, and they live in a high crime area, they may refuse to leave the residence and engage with others for fear of violence. In this case, the intrusiveness of the illness can precipitate social isolation. Another example is a healthy homosexual individual who, due to stigma, experiences minimal social connectedness but does not feel lonely. If this person develops a stigmatizing chronic illness, the additional stigma (or self-stigma) can aggravate the original stigma related to sexual orientation, thus exacerbating the loss of social contacts and sparking feelings of loneliness.

## 4. Outcomes of Social Isolation

Social isolation influences the health outcomes of persons with a chronic illness through a complex, interconnected network of pathways. We classify them as psychological responses, health-related behaviors, and clinical responses.

### 4.1. Psychosocial Responses

The first group of outcomes of social isolation is the psychosocial domain (Figure 1). There is evidence of a significant association between social isolation and depressive symptoms [39]. However, a more in-depth critique of the literature suggests that this evidence is weak [40]. Longitudinal studies have shown that greater loneliness at baseline predicts depression over the subsequent five years [41,42]. Another longitudinal study of more than 1000 young adults [43] found that disconnectedness and loneliness were both associated with depression. However, when entered simultaneously in a regression model, the effect size for loneliness did not substantially change. At the same time, that of social disconnectedness decreased considerably, suggesting that loneliness can be a mediator in the relationship between social disconnectedness and depression. Social isolation is also a well-known risk factor for poor quality of life, for which there is robust literature [44,45].

### 4.2. Health-Related Behaviors

The second group of outcomes of social isolation addresses health-related behaviors or activities performed to promote health and manage chronic illness (Figure 1). These behaviors are consistent with the theory of self-care of chronic illness. According to Riegel, Jaarsma, and Stromberg (2012) [46], people with chronic illnesses perform the following three types of self-care behaviors: self-care maintenance, which includes the healthy practices of regular physical activity, healthy diet, and treatment adherence; self-care monitoring, or the process of observing oneself for signs and symptoms of an illness, and self-care management, or the response to signs and symptoms, such as calling the provider or taking a pill to control a symptom. Self-care is essential in chronic illness to promote health outcomes [47]. However, it has also been found that such behaviors are rarely reported in this population, and one of the reasons for this is the degree of social interactions.

Persons affected by a chronic illness who live alone or have small social networks are more likely to have poor self-care. Evidence to directly support this proposition is lacking; however, we know that older people who are socially isolated are more likely to eat a poor diet and less likely to adhere to regular physical activity [48] than those who are not socially isolated. The reason may lie in the fact that one’s social network both increases the likelihood of receiving support for healthcare, as well as peer pressure to engage in health-promoting practices [49].

Self-care is also negatively associated with subjective social isolation (i.e., loneliness); for example, lonely people have been found to exhibit eating disorders, be more likely to smoke [50], and inconsistent in taking prescribed medications compared to those who are not lonely [51]. We already know that loneliness inhibits socialization; however, the effect of loneliness on self-care behaviors may also be due to a compromised self-regulation of emotion, which diminishes the likelihood of specific lifestyle behaviors such as physical activity [52].

### 4.3. Clinical Responses

We propose that the psychosocial and behavioral effects of social isolation may be determinants of clinical responses, conceptualized in this theory as the third group of outcomes (Figure 1). Accumulating evidence has shown that objective social isolation (i.e., social disconnectedness) [53] and loneliness negatively affect cognitive function [54]. One theory that has been proposed to explain this association is the “use it or lose it” theory, which postulates that intellectual, physical, and social activities stimulate the brain; a lack of participation in social activities results in a decrease in the use of mental faculties, thus explaining the cognitive decline [55].

Cardiovascular diseases (e.g., hypertension, heart failure, and stroke) are also prevalent in lonely and isolated people. The possible mechanisms are related to neuroendocrine dysregulation and hyperactivity of the sympathetic nervous system, leading to hypertension and inflammatory responses [56,57].

Finally, it is well known that socially disconnected and lonely people make greater use of healthcare services [58]. The reason may be attributable both to their poorer health status and a lack of perceived social support, which increases the need for formal healthcare providers to help in the case of health needs (especially emergency department visits).

## 5. Patterns of Social Isolation in Chronic Illness

In this section, we describe four possible patterns of social isolation in the context of chronic illness (Figure 2). For the sake of simplicity, these configurations are presented by considering the two related dimensions of social disconnectedness and loneliness overall, thus leaving interested researchers to investigate each individual indicator of the construct.

### 5.1. Low or Absent Social Disconnectedness and Loneliness

There are scenarios where both the precipitating and predisposing factors are absent or minimal. In this situation, people are more likely to be younger, have good social skills, live in a favorable environment with many opportunities for socialization outside the home, and be affected by a chronic illness with a low level of intrusiveness (e.g., asymptomatic) and not stigmatized. They are also more likely to be surrounded by people that provide emotional and instrumental social support and not suffer from sensory deficits. A typical example is a middle-aged individual affected by essential hypertension that is effectively controlled by a medication regimen and who has strong and active relationships with friends and family members.

### 5.2. Increased Social Disconnectedness and Low or Absent Loneliness

This situation occurs when one’s social network is reduced in terms of the number of interactions and relationship types, but the person does not suffer from loneliness. Drawing on the socio-emotional selectivity theory of Carstensen, Isaacowitz, and Charles (1999) [59], we postulate that the presence of an intrusive chronic illness, in parallel with the process of aging, sparks a progressive selectivity process, in which individuals become increasingly aware of a limited time horizon. Consequently, they invest more in relationships that are emotionally rewarding and supportive (e.g., family members and relatives) and minimize contacts that will not pay off in the future (e.g., non-kin social partners) [60]. The mobilization of intimate helpers in the context of a chronic illness reinforces the feeling of being loved and respected and increases awareness that tangible aid is available in times of need. This form of intimate social support strengthens resilience and the ability to cope with possible stressors (e.g., stigma and ageism), thus reducing the likelihood of loneliness. However, those in this group remain vulnerable because fewer social network members may directly contribute to worse health outcomes through a direct effect (Figure 3).

### 5.3. Increased Social Disconnectedness and High Loneliness

This scenario is exemplified by the individual with a small social network who experiences loneliness. We draw from the evolutionary mechanism for loneliness [61] to describe how people with chronic illnesses are more likely to experience social disconnectedness and feelings of loneliness than healthy individuals. The presence of a chronic illness makes the person more aware of their greater need for emotional and instrumental support (especially when the illness is intrusive) and the increased threat to safety when social contacts are unavailable. This feeling of threat sparks feelings of loneliness, which represents an adverse but evolutionary adaptive reaction, similar to thirst and anger, to re-establish a safe social environment. We postulate that having few social contacts is more common in individuals with poor social skills (i.e., those who have conflictive and poor emotional bonds and those who struggle to maintain healthy relationships) [62]. We assume that the process can also be ignited in the case of a sudden adverse emotional event (e.g., the loss of a family caregiver or a spouse) or a highly stigmatizing illness. In this group, loneliness can have a direct effect on health outcomes or be a mediator of the effect of social disconnectedness on outcomes (Figure 3).

### 5.4. Low/Absent Social Disconnectedness and Increased Loneliness

Some people can be lonely without feeling socially isolated. This is the case of those with strong family and non-family bonds who experience a sudden event connected to the chronic illness, such as the death of an informal caregiver. The course of bereavement elapses without complications, and the person escapes the suffering of grief by interacting with their usual social members. Another case is when the chronically ill person perceives their self-rated health to be poor, which is likely to lead to social loneliness because they misperceive that they can no longer interact with social members in the desired way [63]. In this group, loneliness exposes the subjects to poor health outcomes through a direct effect (Figure 3).

## 6. Assumptions

Assumptions are statements accepted as truth without proof [64]. This theory includes the following four assumptions.

Human beings have an innate desire to interact with others. This premise is based on the concept that humans have an inherent social nature and rely heavily on social contacts to survive and prolong their existence [65].

(1)Chronic illnesses hinder human beings from engaging in social interactions. A vast body of research describes chronic illnesses as disrupting social events [6,66].(2)Loneliness is a traumatic and detrimental form of social isolation, given the psychological pain and distressing state resulting from the experience [67].(3)Loneliness is an experience that people do not seek voluntarily. This assumption is in contrast to objective isolation, which can be manipulated to regulate social adjustment, for example, using social network selectivity [60].

## 7. Propositions

Testable predictions or propositions are part of scientific theories. We propose the following eight testable propositions associated with this theory of social isolation in chronic illness:(1)Higher levels of chronic illness intrusiveness impede social participation and reduce the size of social networks.(2)Social isolation decreases self-care behaviors in people with chronic illnesses.(3)Stigma related to the chronic illness undermines social interactions and predisposes people to loneliness.(4)Social disconnectedness and loneliness in chronic illness patients significantly increase health service use.(5)In chronic illness, the precipitating factors act as triggers to generate social isolation, especially when they occur in clusters.(6)Social network selectivity in chronically ill people protects against loneliness.(7)When an illness is not intrusive and predisposing, and the precipitating factors are absent or minimal, individuals are likely to be socially healthy.

## 8. Clinical and Research Implications

The primary objective of this paper was to present an inductive middle-range theory to describe how the complex phenomenon of social isolation develops during the chronic illness trajectory. We theorized possible predictors and outcomes of social isolation, which offer potential targets for tailored interventions to promote social interactions and minimize the impact of loneliness. Unfortunately, most of the interventional studies conducted to date have targeted older individuals in specific settings (e.g., primary care) or the general community, while relatively few interventions were conducted on the basis of precipitating factors conditioned by the chronic illness [16]. For example, Ellis et al., (2021) [68] reviewed papers that describe the impact of hearing interventions and concluded that the evidence to support their use to treat social isolation is inadequate and insufficient. Other possible interventions were described in order to reduce stigma associated with specific conditions and promote peer interaction, but these issues have received little research attention [69].

Overall, many implications arise from this work. First and foremost, this theory can guide clinical practice; nurses and other health professionals caring for chronically ill individuals should promote screening processes with valid and reliable instruments to understand the extent of isolation and the factors contributing to this phenomenon. Tailored preventative interventions should be designed for at-risk individuals to suppress or limit the impact of the precipitating factors and ultimately promote social integration.

From a research perspective, this theory can be empirically tested, due to the relational propositions formulated. In particular, we suggest that this framework is tested on specific chronic illnesses because living with a chronic illness is a highly subjective experience that involves a delicate adaptation and adjustment process [70], which can precipitate unique, different experiences of loneliness and social disconnectedness.

## 9. Conclusions

Social isolation and chronic illnesses represent two major public health problems whose management has become a primary driver of health and social policies, and nursing care worldwide. This middle-range theory facilitates the evaluation of the construct of social isolation, alongside its predisposing and precipitating factors. Nurses and other healthcare professionals can use this framework to screen for isolation and possibly tailor and test effective interventions to promote social engagement and prevent loneliness.

## Figures and Tables

**Figure 1 ijerph-20-04940-f001:**
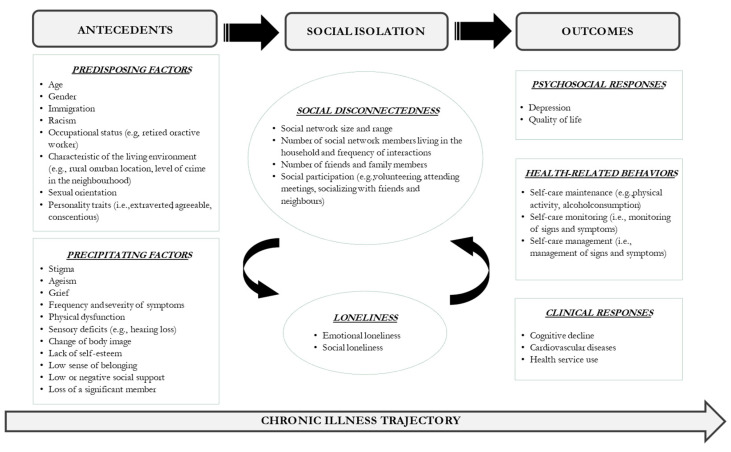
Pictorial model of the middle-range theory of social isolation in chronic illness.

**Figure 2 ijerph-20-04940-f002:**
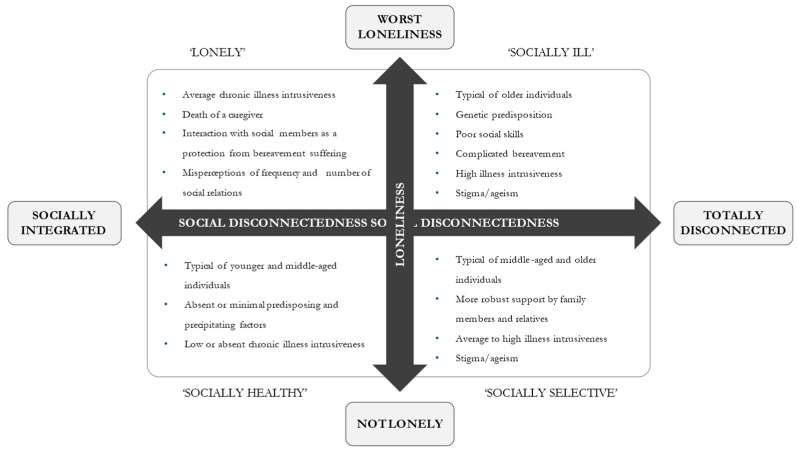
Hypothesized patterns of social isolation and their characteristics in chronic illness.

**Figure 3 ijerph-20-04940-f003:**
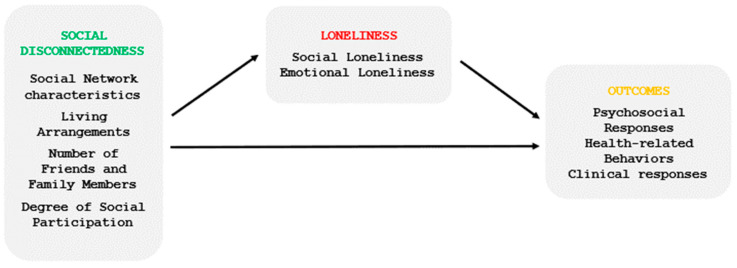
Pictorial representation of the direct and mediating effects of social disconnectedness on health outcomes.

**Table 1 ijerph-20-04940-t001:** Domains and description of the indicators of social isolation (adapted from Cornwell and Waite (2009) [12].

Measure	Domain	Indicator/Description
Social disconnectedness	Social network characteristics	Social network size
		Social network range
		Amount of social network members
		Average frequency of interaction with network members
		Average closeness with network members
	Living arrangements	Household size
		Living alone
	Number of friends and family members	Spouse or current partner
		Number of friends
		Number of children
		Number of grandchildren
	Social participation	Attending religious services
		Attending meetings of an organized group
		Socializing with friends and relatives
		Socializing with neighbors
		Volunteering activities
Loneliness	Emotional loneliness	Lack of an attachment figure to rely on
	Social loneliness	Lack of a larger social network

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
