# Peer review of "A Middle-Range Theory of Social Isolation in Chronic Illness"

_ijerph, 2023, doi:10.3390/ijerph20064940_

Round 1

Reviewer 1 Report

While this is an important paper and presents a theoretical model,It could be refined to be clearer about the social construction of loneliness through patterns of discrimination(not age per se but ageism) and racism as predisposing factors.  Additionally and precipitating factors would include more robust discussion on the elements that are stigma producing (such as blame for one's chronic illnesses) and how our health care environments perpetuate significant disconnected environments for people living with chronic health conditions.  An important contribution, thank you.

Author Response

We sincerely thank the reviewer for raising these important aspects related to social isolation, that we were missing. Ageism is a variable that can explain the degree of social isolation significantly better than the age per se. And the same holds for racism, that can explain loneliness much better than immigration.

We believe that we have addressed all these topics, also including the healthcare environments. We report here the text added to the manuscript:

Explanation of ageism

“Ageism is another precipitating factor for isolation, given that it is relatively similar to stigma. Ageism is defined as the negative stereotypes, prejudices and discriminations toward old age and the aging process. Although it is still a relatively understudied concept, ageism can be an important risk factor for late-life loneliness, through a mechanism of social rejection (e.g., mandatory retirement) and stereotype embodiment (i.e., negative self-perception of aging due to stereotypes) (Avalon et al., 2018). The resultant isolation can be worsened in the presence of an intrusive illness that leads to deterioration of physical health (e.g., compromised mobility)”.

Explanation of racism

“Importantly, this population is also more likely to be exposed to discrimination and racism, which trigger personal insecurity towards social interactions and social participation (Henssler et al., 2020). Racism is a problem not confined to immigration; extant literature suggests that this phenomenon also affects White individuals, with serious consequences, including emotional reactions and feelings of loneliness (Keum et al., 2022).”.

Explanation of general and healthcare environments

Different living environments can impact the lives of people affected by chronic illnesses; for example, a specific health issue can exacerbate feelings of concern about living in a high-crime neighborhood, making them more reluctant to leave their residence. Another factor exacerbating social isolation may be no longer driving due to a decline in physical health. This can be an important issue for people who live in places with few transportation options (National Academies of Sciences & Medicine, 2020). Another factor influencing social isolation is the healthcare environment itself. For example, long-term care residences can increase or decrease isolation, depending on factors such as the provision of home-like accommodations, ease of contact with family and friends, presence of technology, and comfortable private spaces (Andrew et al., 2018)”.

Reviewer 2 Report

Dear Author(s),

social isolation is a major public health challenge, especially for the chronically ill, and that was something that made me read this article. I am aware that it is a concept article, but it lacks clear evidence of implementation of the findings you present.

Kind regards

Author Response

We sincerely thank the reviewer for raising these important aspects related to the manuscript.

After careful discussion with the research team, we decided to add additional content in the clinical and research implications, where we believe it fits best. We hope this approach sufficiently addresses your concerns.

“We theorized possible predictors and outcomes of social isolation, which offer potential targets for tailored interventions to promote social interactions and minimize the im-pact of loneliness. Unfortunately, most of the interventional studies conducted to date have targeted older individuals in specific settings (e.g., primary care) or the general community, while relatively few interventions were conducted on the basis of precipitating factors conditioned by the chronic illness (National Academies of Sciences & Medicine, 2020). For example, Ellis et al., (2021) reviewed papers describing the impact of hearing interventions and concluded that the evidence to support their use to treat social isolation is inadequate and insufficient. Other possible interventions were described in order to reduce stigma associated with specific health conditions and promote peer interaction, but these issues have received little research attention (Thomas et al., 2015)”.

Adapt the keywords according to MESH.

Thanks for raising this point. The keywords are now all MESH terms:

Chronic disease; social isolation; social interaction; loneliness; nursing theory

More than 30% of the references are older than 5 years.

We replaced the older references with more recent ones, which are listed also below. In this way, references older than 5 years have become less than 30%.

  1. Whitehead, L., Jacob, E., Towell, A., Abu‐qamar, M. E., & Cole‐Heath, A. (2018). The role of the family in supporting the self‐management of chronic conditions: A qualitative systematic review. Journal of clinical nursing, 27(1-2), 22-30.
  2. Albert, I. (2021). Perceived loneliness and the role of cultural and intergenerational belonging: The case of Portuguese first-generation immigrants in Luxembourg. European Journal of Ageing, 18(3), 299-310.
  3. Iveniuk, J. (2019). Social networks, role-relationships, and personality in older adulthood. The Journals of Gerontology: Series B, 74(5), 815-826.
  4. Welch, L., Sadler, E., Austin, A., & Rogers, A. (2021). Social network participation towards enactment of self‐care in people with chronic obstructive pulmonary disease: A qualitative meta‐ Health Expectations, 24(6), 1995-2012.
  5. Lara, E., Caballero, F. F., Rico‐Uribe, L. A., Olaya, B., Haro, J. M., Ayuso‐Mateos, J. L., & Miret, M. (2019). Are loneliness and social isolation associated with cognitive decline?. International journal of geriatric psychiatry, 34(11), 1613-1622.
  6. Leigh-Hunt, N., Bagguley, D., Bash, K., Turner, V., Turnbull, S., Valtorta, N., & Caan, W. (2017). An overview of systematic reviews on the public health consequences of social isolation and loneliness. Public health, 152, 157-171.
  7. Stachteas, P., Symvoulakis, M., Tsapas, A., & Smyrnakis, E. (2022). The impact of the COVID-19 pandemic on the management of patients with chronic diseases in Primary Health Care. Population Medicine, 4(August), 1-13.
  8. Polenick, C. A., Perbix, E. A., Salwi, S. M., Maust, D. T., Birditt, K. S., & Brooks, J. M. (2021). Loneliness during the COVID-19 pandemic among older adults with chronic conditions. Journal of Applied Gerontology, 40(8), 804-813
  9. Thomas, N., McLeod, B., Jones, N., & Abbott, J. A. (2015). Developing internet interventions to target the individual impact of stigma in health conditions. Internet Interventions, 2(3), 351-358.

Please, improve the conclusion section.

See above.